# Water quality improvements offset the climatic debt for stream macroinvertebrates over twenty years

Ian P. Vaughan [1,2] & Nicholas J. Gotelli[2]

Many species are accumulating climatic debt as they fail to keep pace with increasing global temperatures. In theory, concomitant decreases in other stressors (e.g. pollution, fragmentation) could offset some warming effects, paying climatic debt with accrued environmental credit. This process may be occurring in many western European rivers. We fit a Markov chain model to ~20,000 macroinvertebrate samples from England and Wales, and demonstrate that despite large temperature increases 1991–2011, macroinvertebrate communities remained close to their predicted equilibrium with environmental conditions. Using a novel analysis of multiple stressors, an accumulated climatic debt of 0.64 (±0.13 standard error) °C of warming was paid by a water-quality credit equivalent to 0.89 (±0.04)°C of cooling. Although there is finite scope for mitigating additional climate warming in this way, water quality improvements appear to have offset recent temperature increases, and the concept of environmental credit may be a useful tool for communicating climate offsetting.

[1] Cardiff School of Biosciences and Water Research Institute, Cardiff University, Cardiff CF10 3AX, UK. [2] Department of Biology, University of Vermont, Burlington, VT 05405, USA. Correspondence and requests for materials should be addressed to I.P.V. (email: VaughanIP@cardiff.ac.uk)

In response to global climate change, species will either shift in distribution as they track their preferred climate, adapt locally, or decline in abundance ultimately to extinction[1,2]. A useful concept for understanding these changes is climatic debt[3,4], which was inspired by the idea of extinction debt[5]. Assuming that species' abundances and geographic ranges will reach an equilibrium in a constant environment, the environmental lag (sensu ref. [6]) describes how far the observed community is from the equilibrium predicted by current environmental conditions. In the case of temperature and climatic debt, this lag relates to communities responding more slowly than expected to rising temperatures[4]. Such debts have been documented for a range of taxa including forest plants, birds, and butterflies[3,7–9], and must eventually be paid through migration, local adaptation, or extinction.

The climatic debt model has focused on warming as the sole environmental determinant of community change. However, recent work has shown how a range of biotic and abiotic factors contribute to the total debt[4]. In theory, climatic debts could be reduced by management strategies such as assisted migration to unoccupied sites[10] or the introduction of genotypes that are better adapted to warmer conditions[11].

An important alternative to the climatic debt model is that communities are not in disequilibrium with environmental conditions. Instead, communities are in equilibrium, but are affected simultaneously by changes in temperature and other environmental factors, which could either compound or offset the effects of rising temperature[12]. Reducing a second stressor accumulates environmental credit which could be used to pay the climatic debt, and may be an important mitigation strategy to limit damage from increasing temperatures.

Here, we introduce the novel concept of water quality credit, which may have paid the climatic debt accumulated by macroinvertebrate communities in English and Welsh rivers through the 1990s and early 2000s. Riverine macroinvertebrates are sensitive to climate[13], and water temperature increases equivalent to those observed in the current study often account for large changes in community structure (e.g. refs. [14,15]). Concomitantly, as in much of Western Europe, water quality in British rivers has improved greatly in recent decades due to better treatment and reduced discharge of industrial and sewage effluent[16]. This has prompted large-scale biological recovery that may have offset concurrent water temperature increases[17–19] (Supplementary Fig. 1).

Across a series of analyses using ~20–26,000 high-quality, standardised benthic macroinvertebrate samples collected from 3067 stream locations across England and Wales (Supplementary Fig. 2), we estimate the overall environmental lag of riverine communities, and the contributions of climatic debt and water quality credit that accumulated through the period (Supplementary Fig. 3). Our results indicate that, at a national scale, macroinvertebrates responded rapidly to changes in water quality and temperature, remaining close to equilibrium with the environment throughout the study period. Because water quality credit paid the accumulating climatic debt, the overall environmental lag was small.

## Results

**Macroinvertebrate community classification.** We first created a simple classification of riverine macroinvertebrate communities based upon their constituent taxa. The classification divided communities into three types, representing an ordered sequence from communities characterised by pollution-sensitive taxa requiring fast-flowing, well-oxygenated water (e.g. Plectopera, Ephemeroptera; Class 1) to communities characterised by pollution-tolerant taxa that can persist in slow-flowing, poorly oxygenated water (e.g. Hirudinea, Isopoda; Class 3; Fig. 1a and Supplementary Fig. 4a). Similar results were obtained with different classification methods (Supplementary Fig. 4b).

**Markov chain modelling.** The classification was used to construct a time-varying Markov transition model that described annual changes in river community types through time (Fig. 2) and then allowed us to assess whether the dynamical stability of macroinvertebrate communities had changed during the study period. Macroinvertebrate communities were dynamic, with approximately 20% of communities switching classes annually (Supplementary Fig. 5a; Supplementary Table 3) and long-term trends in the transition probabilities were consistent with national-scale improvements in water quality[18]: the annual persistence of polluted assemblages (Class 3) decreased through time (Fig. 2), leading to a 50% decrease in the frequency of polluted assemblages and an increase in the frequency of unpolluted and intermediate assemblages (Classes 1 and 2; Fig. 1b, Supplementary Table 4). The prevalence of Classes 1 and 2 increased at similar rates (Fig. 1), but slightly faster for Class 2, driven by increasing probabilities through time of switching from Class 3 to Class 2, decreasing probabilities of switching from Classes 2 to 1,

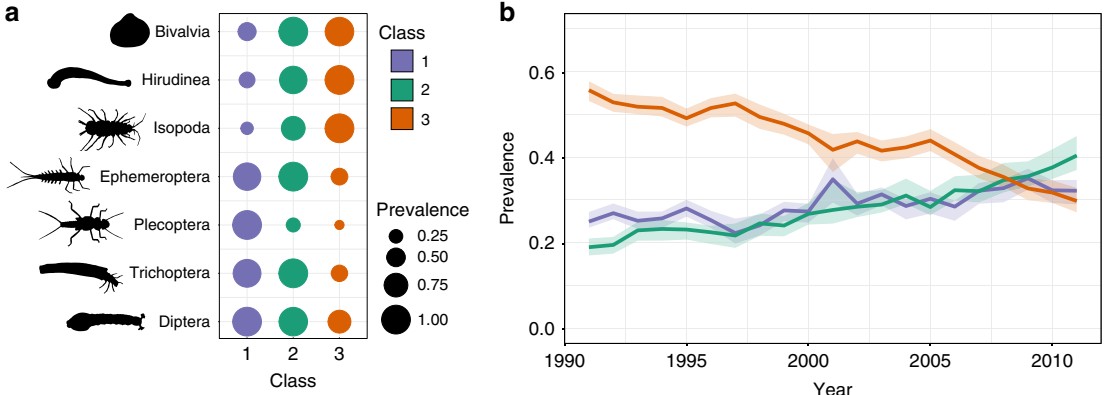

**Fig. 1** Macroinvertebrate community composition and change across England and Wales 1991–2011. **a** Characteristic changes in the community among the three biological classes: the mean prevalence of families within the seven higher taxa showing the greatest change among classes, scaled so that the relative prevalence of each taxon = 1 in the class where it was most prevalent, and **b** the change in prevalence of the three classes, with bootstrapped 95% confidence limits. See Supplementary Fig. 4a for a full version of **a**. Source data are provided as a Source Data file. Invertebrate silhouettes drawn by I. Vaughan

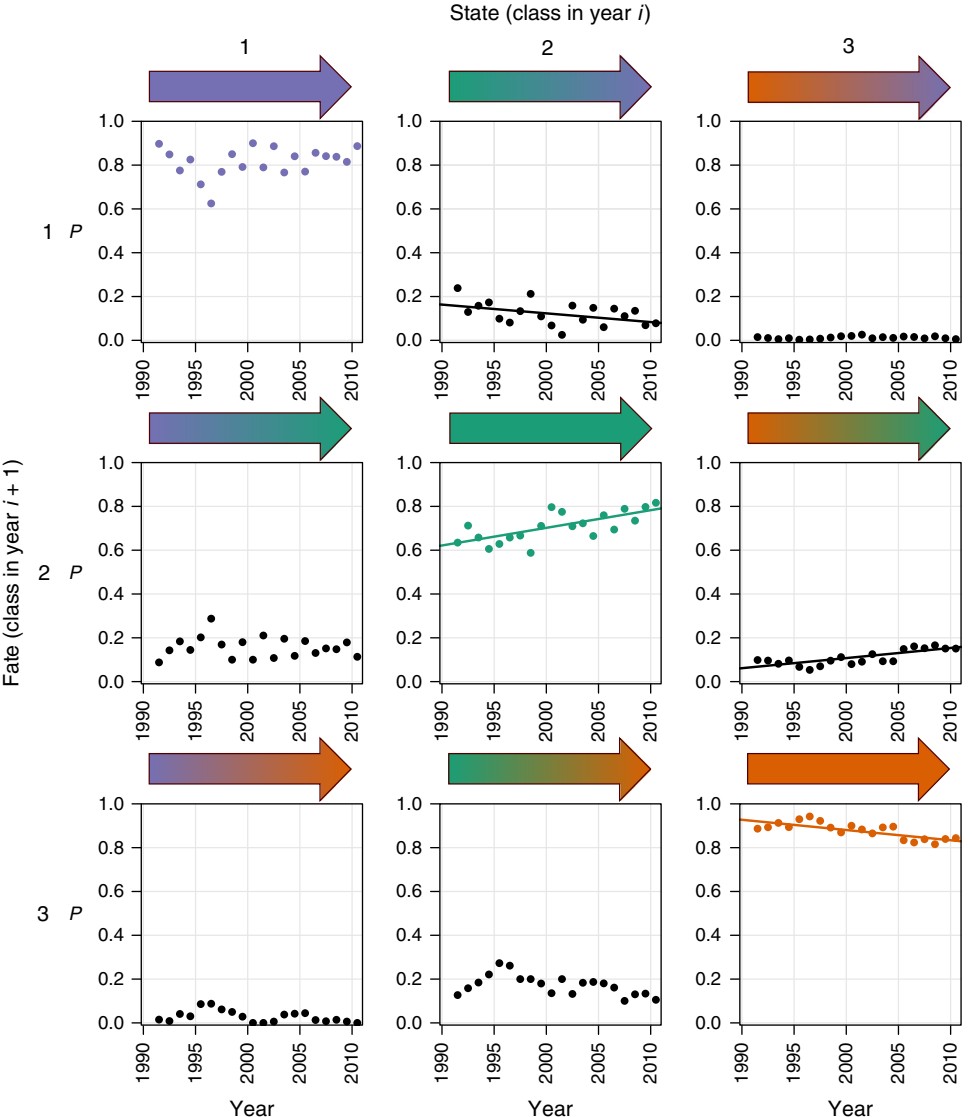

**Fig. 2** Annual transition probabilities between the three macroinvertebrate community classes (1991–2011) across England and Wales, based on the Markov chain. Each panel shows the probability (P) of a community in one of the three states in 1 year either staying in the same state or switching to one of the other two in the following year (its fate). Colours follow Fig. 1, with fill colours of the arrows indicating the transitions between the different states. The three panels forming the leading diagonal represent the probability of communities staying in the same state between years (= the persistence). Fitted linear regression lines are shown where the slope coefficient is significant at $P \leq 0.05$ (see Supplementary Table 4 for regression results). Source data are provided as a Source Data file

and increased persistence of communities within Class 2 (Fig. 2). Classes 1 and 2 expanded their geographic distributions, while Class 3 retreated mainly to urban areas and slow-flowing rivers in the intensively farmed landscapes of eastern England (Fig. 3; Supplementary Fig. 2a). Class 1 communities were concentrated in northern and western areas, where fast-flowing, well-oxygenated rivers are most abundant[18], whereas Class 2 expanded across the lowlands of England (Fig. 3; Supplementary Fig. 2a).

Despite the changes in composition and transition probabilities, the dynamical properties of the fitted Markov model changed little at the national scale. Asymptotic (damping coefficient) and non-equilibrium (Dobrushin's coefficient) dynamical stability measures both indicated a recovery rate from perturbation of 15–20% year$^{-1}$ (Supplementary Fig. 5a). There are few published examples of similar analyses, but these recovery rates are comparable to those estimated for sub-tidal benthic communities[20] and indicate a resilient system that can recover quickly from perturbation or respond rapidly to environmental change. These stability results were not sensitive to how the invertebrate communities were clustered prior to fitting the Markov chain model (Supplementary Fig. 5b).

The overall community composition (prevalence of Classes 1–3) was close to the estimated equilibrium state throughout the monitoring period (Supplementary Fig. 6), suggesting that there was minimal net environmental lag. Collectively, these results indicate that monitored changes in the macroinvertebrate community across England and Wales in recent decades[18] represent a response to contemporaneous environmental change, rather than a time-lagged response to past changes, reinforcing their value for environmental monitoring. In this scenario, macroinvertebrates respond to environmental change in a relatively predictable fashion[6], supporting efforts to model climate change impacts (e.g. refs. [21,22]).

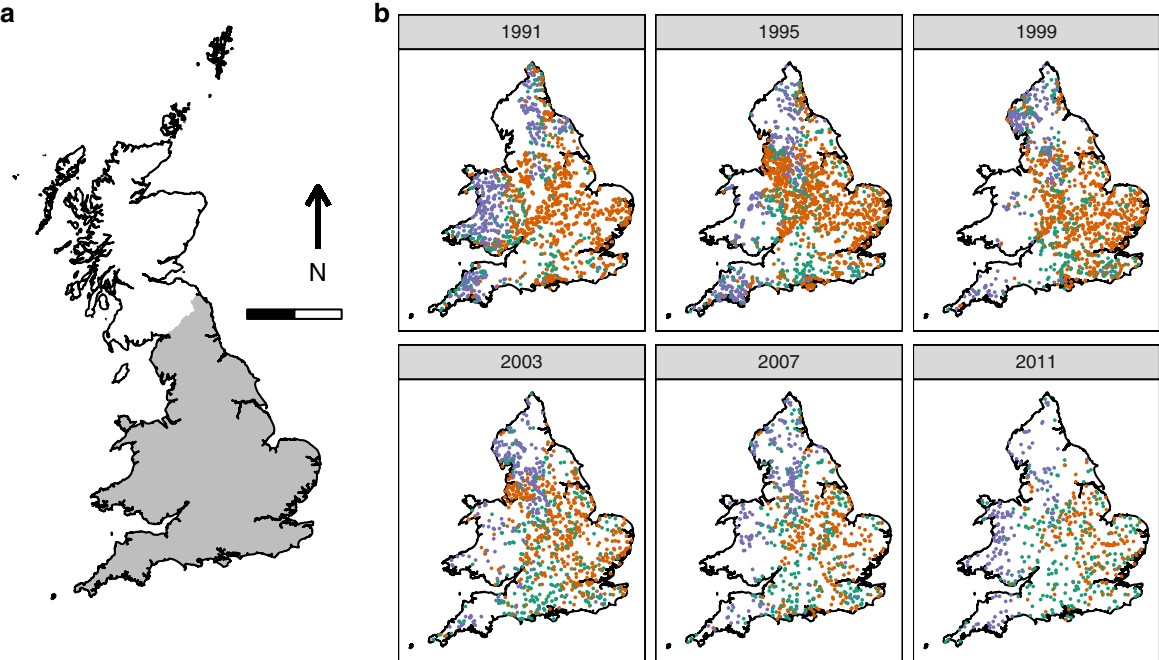

**Fig. 3** The study area and locations of macroinvertebrate samples collected for calculating transition probabilities among years. **a** Outline of Great Britain, indicating England and Wales (shaded): the scale bar is 200 km. **b** The samples collected in each year (sample sizes are given in Supplementary Table 1), shaded according the biological class to which they belonged. The colour scheme matches Fig. 1. For brevity, data are shown for four-year intervals (see Supplementary Fig. 2 for a full version). National outlines contain OS data © Crown copyright and database right (2017)

**Climatic debt and water quality credit**. Although the Markov model revealed minimal net environmental lag in English and Welsh rivers, this does not preclude the formation of debts and credits that may have cancelled each other out, leading to a negligible net lag. Specifically, a water quality credit may have formed if decreases in pollutant concentrations could negate increases in water temperature. To investigate this possibility, we focused on the biochemical oxygen demand (BOD), which provides an overall assessment of organic pollution[16], reflects one of the largest water quality improvements in recent decades[18] (Supplementary Fig. 1), and impacts aquatic organisms in similar ways to increasing water temperature: both promote oxygen stress to meet respiratory demand, and increased oxygen availability can reduce the sensitivity of freshwater macroinvertebrates to increasing water temperature[13,23,24]. Also, compared to point measurements of dissolved oxygen, BOD better reflects the oxygen conditions that macroinvertebrates experience in their benthic microhabitats (e.g. ref. [23]). Using data from 1991 to 1992, we calibrated models that predicted temperature and BOD from the composition of the invertebrate assemblage, and used them to reconstruct how these two variables were expected to change based on the observed changes in the invertebrate community between 1991 and 2011 (Supplementary Fig. 3; ref. [3]). The differences between reconstructed and observed conditions represent environmental lags[6] (Fig. 4a–c).

Although BOD and temperature were weakly correlated in the observed data ($r = 0.05$, $n = 26,103$), their reconstructed values were highly correlated ($r = 0.74$), consistent with a similar biological response to increasing temperature or BOD. Over the monitoring period, observed mean water temperature increased, whereas reconstructed temperature decreased—consistent with the observed increase in the prevalence of taxa preferring cooler conditions[19] (i.e. Class 1)—generating an estimated climatic debt of 0.64 °C by 2011 (Fig. 4d; Supplementary Table 5). Concomitantly, observed BOD declined faster than the reconstructed BOD, leading to an estimated water quality credit of 0.63 mg l$^{-1}$

(back-transformed value; Fig. 4e; Supplementary Table 5). We cross-calibrated reconstructed BOD and temperature so that the change in BOD could be converted to an equivalent temperature change (Supplementary Fig. 7). This calibration yielded a water quality credit equivalent to 0.89 °C, paying the climatic debt and potentially leaving a small net water quality credit, although this credit did not differ significantly from zero (linear regression: $P = 0.06$; Supplementary Table 5; Fig. 4f).

## Discussion
In response to climatic warming, management that reduces other stressors at local or regional scales may be an effective way of reducing ecological impacts and increasing resilience[12]. Such interventions are seen to be more achievable than change at the global scale[12], and may provide more time for species and ecosystems to adapt to climate change[25]. This approach can also provide secondary biodiversity and ecosystem service benefits, such as reducing risks of harmful cyanobacterial blooms and lowering drinking water treatment costs[26,27]. Because most ecosystems are subject to multiple stressors[28], the success of offsetting global change in this way relies upon an understanding of how different stressors interact[12]. If local stressors such as water quality have additive or synergistic effects with climate, reducing their impacts could be a valuable mitigation tool. However if local stressors act antagonistically with climate, the potential for offsetting will be limited[25].

Our credit–debt framework applies where the ecological impacts of different stressors can be cross-calibrated, allowing their actions to be expressed on a common scale (°C in this study). This framework would readily generalise to three or more stressors, allowing more complete accounting of the credits and debts within a system, using a common currency to facilitate stakeholder engagement. In the case of rivers, basin-scale management could compare the potential credits accrued though different strategies such as reducing point source or diffuse

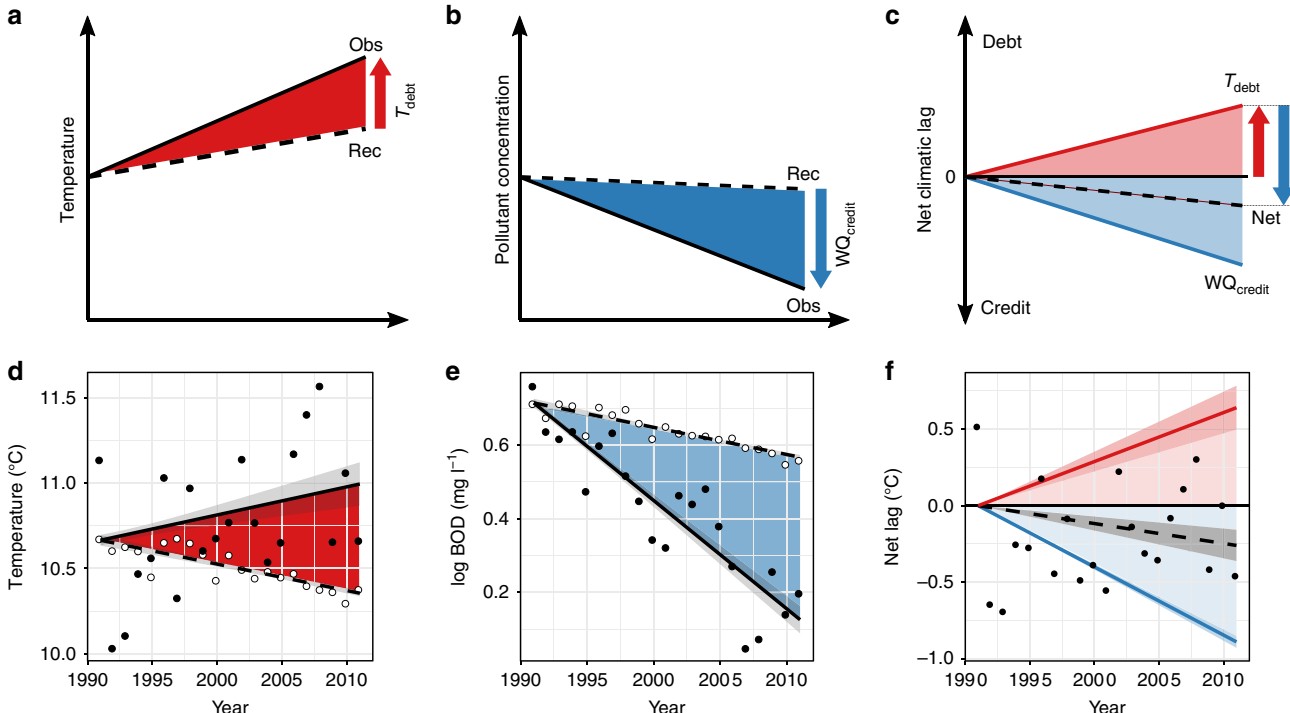

**Fig. 4** The concepts of climate debt, water quality credit, and the net environmental lag, and the estimated values for English and Welsh rivers 1991–2011. **a** An increase in the observed (Obs) water temperature, not matched by an increase in the reconstructed (Rec) temperature, leads to a climatic debt accumulating ($T_{debt}$). **b** A decrease in the observed pollutant load that is not fully matched by an equivalent change in the community, hence by the reconstructed pollutant load, generates water quality credit ($WQ_{credit}$). In **c** water quality credit has been converted to an equivalent change in temperature (see Supplementary Fig. 7), and the increase in water quality credit (blue) and climatic debt (red) are plotted against time, along with the net climatic lag (black dashed line). In this example, credit > debt so that the net change is negative, representing a small water quality credit, despite rising temperatures. **d–f** The observed accumulation of climatic debt (**d**) and water quality credit for BOD (**e**) across England and Wales; filled symbols represent the observed conditions; open symbols the reconstructed conditions. **f** Climatic debt, water quality credit, and the net lag (filled points; black, dashed line). Regression lines fitted using generalised least squares, with bootstrapped standard errors (see Supplementary Table 5 for full outputs of the regression models). Source data are provided as a Source Data file

pollutant inputs, including fine sediments, and expanding riparian tree cover to shade the channel. In the same way as for biodiversity offsetting[29], the value of the credit–debt approach depends in part upon how closely the reduction in one stressor counteracts an increase in another. While closely related via their impacts on oxygen stress, changes in temperature and BOD will not have identical ecological effects, so offsetting will inevitably be imperfect. This limitation is likely to increase where the mechanistic link between stressors is weaker.

In the UK, it is difficult to judge how far into the future the benefits of water quality credit may extend. By 2011, mean BOD was 1.1 mg l⁻¹, below the reference threshold of 2.4 mg l⁻¹ for much of Western Europe[30], although BOD below this level still affects invertebrate abundance[31]. Further reductions in BOD may have limited capacity to offset increasing temperatures, and we predict that further warming will lead to a reversal of the trends of the past two decades and an increase in the persistence and frequency of Class 3 polluted assemblages.

The Markov model and estimation of environmental lags provide independent lines of evidence that benthic macroinvertebrate communities were close to equilibrium at the national scale, despite large environmental and biological changes. The potential for water quality to offset rising temperatures is consistent with both experimental (e.g. ref. [23]) and observational (e.g. refs. [17,32]) studies, but this is the first time that the debt and credit have been quantified and directly compared in the field. The expanded framework of climatic debts and environmental credits can be applied to other systems in which reductions in

environmental stressors could, within limits, potentially mitigate rising temperatures. Using common units to compare debts and credits (°C) could make this a valuable tool for communicating mitigation options among scientists, stakeholders, and policy makers.

## Methods
**Overview of the methods.** There were three parts to the study (Supplementary Fig. 3): (i) collation of national-scale data sets for macroinvertebrate communities, water chemistry, temperature, and discharge; (ii) classification of macroinvertebrate communities into three types and specification of a Markov chain transition model based on the probabilities of community state change through time. This model was used to quantify how close communities were to equilibrium with the environment (i.e. whether an environmental lag was present); (iii) comparison of observed long-term changes in abiotic conditions (primarily water temperature and BOD) with those predicted from changes observed in community structure, allowing an assessment of environmental debts, credits, and net lags. All data analyses were carried out in R version 3.4 (ref. [33]) and the fit of regression models checked using plots of the residuals[34].

**Collation of biological and physico-chemical data.** Macroinvertebrate data were collected by the Environment Agency in the course of routine monitoring of rivers across England and Wales over a period of many years. Locations were selected from the Agency's database if they: (i) had at least one instance in which they were sampled in consecutive years so that annual transition probabilities could be estimated for the Markov state transition model, below, and (ii) included the elevation, channel slope, and distance from source of the location, as these are effective proxies for physical habitat[35].

To minimise spatial autocorrelation, we selected the most frequently sampled location from each Water Framework Directive waterbody[36], leading to 3067 locations (Supplementary Figs. 2 and 3). Our analyses focussed upon 1991–2011 because: (i) this was a period with good spatial and temporal coverage across

England and Wales, and (ii) an extensive quality assurance procedure revealed a near-constant error rate through time (see ref. [18]). Sampling intensity varied through space and time (mean = 948 locations sampled per year; Supplementary Fig. 2a), but a similar subset of rivers was sampled in each year (Supplementary Fig. 2b). Where a location was sampled multiple times in the same year, one sample was selected at random. This led to a final sample size of 19,915 samples, creating 14,343 annual transitions: a mean of 6.68 per site or 717.2 per year (Supplementary Table 1).

Macroinvertebrate samples were taken in spring (March–May) using a standard 3-min kick sampling protocol[37]. Identification was primarily to family level: here we follow the taxonomy of Vaughan and Ormerod[19] ($n = 78$ taxa). Taxon abundance was recorded either as counts of individuals or the $\log_{10}$ abundance class to which each taxon belonged (e.g. 1–9 individuals or 10–99 individuals). To harmonise the abundance data, we converted all data to $\log_{10}$ abundance classes and recorded the abundance as the mid-point of that class (e.g. the class 1–9 was recorded as 5.5). The classification and Markov model analyses below were run with both presence–absence and abundance versions of the data: in the main paper we focus upon the conservative presence–absence approach.

Water chemistry and temperature were recorded by the Environment Agency at monthly intervals over the same time period as the biological sampling. In addition to water temperature, we focused on three water quality determinants: BOD, nitrate, and orthophosphate (sampled using standard methods[18]). Total oxidised nitrogen was sometimes recorded in place of nitrate. However, because nitrate often exceeded 99% of the total when both were recorded, we regressed nitrate on total oxidised nitrogen (linear regression: $n = 79{,}781$, $R^2 = 0.98$) and used this to impute missing nitrate values ($n = 20{,}217$; ref. [18]). We filtered the data set to remove locations that were immediately downstream of outfalls from sewage treatment works or industrial discharges.

Median water chemistry and temperature were calculated at each location for the 12 months prior to the spring biological sampling period (i.e. March–February inclusive); years at a sampling site were rejected if they contained fewer than nine monthly samples. This resulted in an initial data set with 10,790 locations (mean of 9.7 years per location; 1991–2011) with all four variables sampled. When >50% of samples were below detection limits, annual medians were imputed at each location using the method of regression-on-order-statistics (NADA package[38]).

Daily mean discharge data were sourced from the Centre for Ecology and Hydrology's National River Flow Archive. We used data from 943 stations across England and Wales (mean = 19.3 years 1991–2011) and calculated annual median discharge for the same 12 month periods as for chemistry. The discharge in $m^3\,s^{-1}$ was divided by the catchment area and expressed in units of $mm\,day^{-1}$ (ref. [39]) to give a standardised run off measurement that was independent of stream size (catchment area).

Some of the biological locations corresponded with the chemistry and hydrology, but most locations were a short distance apart. Fortunately, in most cases chemistry and hydrology samples were collected nearby on the same watercourse (mean distance = 3.8 km). We therefore interpolated water temperature, water chemistry measurements, and discharge in every year using local kriging with a 25 km neighbourhood (gstat package[40]). Measured or imputed physical variables were available for >95% of biological sampling locations in each year (mean = 2980 year$^{-1}$). In all analyses, water chemistry and discharge were log transformed.

### Biological classification
Macroinvertebrate community composition varies along a continuum in English and Welsh rivers[18], so was divided into an arbitrary number of similar-sized classes using cluster analysis. We repeated the analysis with different clustering techniques and numbers of clusters to ensure that the final conclusions were not sensitive to the choices made. We focus upon the simplest approach using three classes because this provided the simplest interpretation and maximised the sample size within each class to produce more consistent estimates of transition probabilities (see below).

The 19,915 samples were classified using a hierarchical clustering method, Ward's method, and a non-hierarchical approach, partitioning around medoids (cluster package[41]). Both methods utilise a dissimilarity matrix, and we calculated Jaccard dissimilarities for presence–absence data and Bray–Curtis dissimilarities on the abundance data. In total, there were 12 permutations: two clustering algorithms, two distance matrices, and three, five, or seven cluster-groupings that were analysed. Results are presented for the combination of Jaccard distance and Ward's clustering, but most analyses were re-run with the other permutations to ensure that the results were not sensitive to this choice.

Temporal trends in the prevalence of the three biological classes across England and Wales were estimated using a generalised additive model (GAM) following Fewster et al.[42]. With the emphasis on annual transitions between classes, we fitted year as a factor, rather than using a smoothing term. Due to the large number ($n = 3067$) of relatively short and sparse time series, we used six site-level covariates instead of fitting site as a factor[18,42]: altitude, slope, distance from source, the proportion of the WFD catchment that had urban land cover (estimated from the UK Land Cover Map 2000; ref. [43]) and the longitude and latitude of the site. The latter two were modelled using a tensor product of thin-plate smoothing splines (mgcv package[44]). All variables except longitude and latitude were log + 1 transformed to improve the fit of the model. Nonparametric bootstrapping was used to estimate 95% confidence limits around the prevalence of each class[42].

### Markov state transition model
To capture dynamical properties of the system we defined a Markov chain based on the annual probabilities of macroinvertebrate communities remaining in the same class, or switching to another class in each time step. Following Caswell[45] and Hill et al.[20], the analysis proceeded in two stages: (i) we used a loglinear analysis to select a suitable model specification, comparing alternative models in which transition probabilities were either fixed or allowed to vary between different river types or through time; and (ii) we then analysed the best-fitting model to calculate the expected frequencies of the different classes at equilibrium, and the dynamic stability of the equilibrium.

Following Caswell[45], we fit a hierarchy of models in which transition probabilities are homogeneous across space and time, vary only in space or time, or are allowed to vary across both space and time (Supplementary Methods). In this study, space was represented by different river types, dividing the rivers into rural and urban, or upland and lowland channels. In every case, transition probabilities varied through time and among river types (loglinear analysis: all $P < 0.001$; Supplementary Table 2a), but temporal differences in transition probabilities among river types were weak: the space–time interaction was only significant (at $P < 0.05$) in three out of eight instances and was strongly rejected by the AIC model comparisons (Supplementary Table 2b). Based on the AIC, the model containing space and time (but no interaction) was the optimal model in every case. Addition of the interaction term greatly reduced the support for the model in all eight cases (delta AIC > 70). These results suggest that while the prevalence of the biological classes differed among river types, the temporal trends were similar. As the focus of the study was primarily temporal (cf. spatial) change, and the mix of upland–lowland and rural–urban rivers sampled was near-constant through time (Supplementary Fig. 2b), the selected model assumed spatial homogeneity but allowed transition probabilities to vary among years ($n = 20$ transitions).

The Markov model estimated the probabilities of macroinvertebrate communities in each class (i.e. their current state) either remaining in the same class or moving to a different class at each transition period (i.e. their fate). Three properties were estimated from the Markov chain for each transition period: (i) the prevalence of the three biological classes at equilibrium (=the stationary community[20]); (ii) the proportion of communities remaining within the same class, referred to as the persistence; and (iii) two measures of the convergence rate of the community following disturbance: the damping ratio, $\rho$, which is the ratio of the dominant eigenvalue to the second largest eigenvalue (popbio package[46]) and Dobrushin's coefficient, $\bar{\alpha}$ (ref. [47]). The damping ratio measures the return time to equilibrium following a small local perturbation, whereas Dobrushin's coefficient measures the rate of convergence of two communities that are in different non-equilibrium states[47]. Return rates (year$^{-1}$) for the two measures are calculated as $\log \rho$ and $-\log \bar{\alpha}$ (ref. [47]). The mean absolute difference between the observed prevalence of the biological classes and their equilibrium prevalence was used as a simple measure of the system's proximity to equilibrium.

Overall trends in transition probabilities and transition matrix statistics through time were modelled using linear regressions fitted with generalised least squares (GLS). In every case, the AIC was used to select between models including first-order moving average and autoregressive error distributions, and a basic model without a term for temporally autocorrelated residuals[34]. In every case, the simplest model within two AIC units of the model with the lowest AIC was selected. Models were fitted using the rms package, with standard errors estimated from 400 bootstraps[48].

To synthesise the temporal changes in transition matrix statistics across all 12 permutations of the cluster analysis (two dissimilarity metrics × two clustering methods × three, five, or seven clusters), we fitted mixed effects models using the nlme package[49]. We fitted random slopes and intercepts, as well as first-order autoregressive or moving average error terms, and selected the most parsimonious model within two AIC units of the smallest AIC. The mean absolute difference between the observed and equilibrium prevalence was log transformed to improve model fit.

### Estimating climatic debt and water quality credit
We estimated climatic debt following Bertrand et al.[8], calibrating transfer functions that allowed temperature or water chemistry to be predicted from the observed macroinvertebrate community. The same approach is well-established in reconstructions of paleoclimates (e.g. refs. [50,51]). Transfer functions were created for water temperature and BOD, the main focus of the debt and credit analyses. We also analysed air temperature and a combination of orthophosphate, nitrate, and discharge, all three of which were highly correlated ($|r| = 0.62$–66; $n = 26{,}103$). For air temperature, we used mean annual values calculated for the same periods as water temperatures, calculated from the UK Meteorological Office[52]. For water chemistry, we used the first component from a principal component analysis containing the three variables (explaining 75% of the variance). In all cases, we refer to the predicted conditions as being reconstructed to distinguish them from direct empirical measures of temperature or chemistry.

All biological samples from the 3067 sampling locations, for which kriged water temperature, chemistry, and discharge were available, were used ($n = 26{,}103$). In lieu of an independent data set for calibrating the transfer function, we used samples from 1991 to 1992 to calibrate the models ($n = 2751$), which were then used to make predictions for the complete data set (1991–2011) to reconstruct the expected changes in the environment. In contrast to previous studies (e.g. ref. [8]), our transfer functions were calibrated during a period of known environmental and

biological change[18], when environmental lags may already have been present, introducing potential bias into the transfer function[6]. We do not think that this was a problem here because the system appeared to be close to equilibrium throughout the study period (see Markov results). At worst, the results should indicate changes in the relative, rather than absolute, environmental lags.

Transfer functions were created using weighted averaging partial least squares[53] (WA-PLS), a well-established approach for historical climate reconstruction. WA-PLS models were fitted with the rioja package[54], using the $\log_{10}$ macroinvertebrate abundance data. Leave-one-out cross-validation was used to select the model complexity (number of components retained) and estimate the overall fit to the 1991–1992 training data[54]. For water temperature, we selected one component (cross-validated $R^2 = 0.15$), and for BOD, three components ($R^2 = 0.43$). Air temperature retained two components ($R^2 = 0.43$) and combined discharge and nutrient concentrations retained three components ($R^2 = 0.64$).

Mean annual observed and reconstructed conditions were calculated across the 3067 locations, and the environmental lags estimated by subtracting the reconstructed conditions from the observed values[4]. Climatic debt is defined as a positive lag for temperature, with the observed temperatures rising faster than the changes in the reconstructed temperature[4]. Extending this analogy, water quality credit was defined as a faster improvement in the observed relative to the reconstructed quality, producing a negative lag.

To allow a direct comparison of climatic and BOD lags, we converted BOD to equivalent predicted water temperatures (Supplementary Fig. 7). We fitted a GAM, regressing reconstructed temperature onto reconstructed BOD ($n = 27,378$) using a penalised thin-plate regression spline, with the degree of smoothing selected using generalised cross-validation[44]. The observed and reconstructed BOD were converted to temperature equivalents using the fitted model and the BOD lag was calculated in degrees Celsius. With the water quality lag expressed in Celsius, the net environmental lag was estimated by summing the climatic and BOD lags. This process was repeated using the discharge and nutrients principal component, and air temperatures in place of water temperatures.

GLS models were used to test for trends in: (i) the observed and reconstructed conditions, leading to estimates of climatic and water quality lags, and (ii) climatic, water quality, and the net environmental lag, following the same protocol as for transition probabilities (above). For the climatic and water quality lags, an interaction term between year and type of time series (observed or reconstructed conditions) was included to allow a different slope to be fitted to each one. Models were fitted to (year – 1991), so that the time series started at zero, and the intercept constrained to be the same for the two time series so that the estimated lag was zero in 1991. In addition to assessing first-order structures for autocorrelated errors, we assessed an error variance structure that allowed heterogeneity of variance among the time series[34]. For the net environmental lags, intercepts were set to zero, so that lags were zero at the start of the time series.

Finally, we re-fitted the GLS models for temperature, water chemistry, and net environmental lags without constraints on the $y$-intercepts to check that this did not strongly influence the results. For the combination of water temperature and BOD, the results were similar, with a smaller net lag estimated compared to when the $y$-intercept was fixed at zero (Supplementary Fig. 8; Supplementary Table 5). Using air in place of water temperatures, a larger net water quality credit was estimated, although still not significantly different from zero (Supplementary Fig. 9). A combination of nutrients and discharge only offset a fraction of the climatic debt (Supplementary Fig. 9), consistent with eutrophication having weaker biological effects than BOD within the range of conditions typically encountered in lowland Europe[31].

**Reporting summary**. Further information on research design is available in the Nature Research Reporting Summary linked to this article.

## Data availability
The source data underlying Figs. 1, 2, 4d–f, Supplementary Figs. 1, 4–6, 8 and 9, and Supplementary Tables 3–5 are provided as a Source Data file. The full data set, including the biological classifications, is available from https://github.com/ivaughan/climatic_debt

## Code availability
The R code to reproduce the analyses is available from https://github.com/ivaughan/climatic_debt.

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

## Acknowledgements

We thank the Environment Agency and Natural Resources Wales for supplying the data. The work was supported by a Cardiff University Research Leave Fellowship awarded to I.P.V.

## Author contributions

I.P.V. and N.J.G. devised the study and wrote the manuscript. I.P.V. analysed the data.

## Additional information

**Competing interests:** The authors declare no competing interests.

