## [Peer Review File · Nature Communications]

Reviewers' Comments:

Reviewer #1:

Remarks to the Author:

The manuscript by Vaughan and Gotelli analyses whether improving water quality in recent decades may be offsetting the ecological effects of climate warming in rivers. Overall, I enjoyed reading this manuscript and think it is a generally important and timely contribution.

The analysis uses an impressive data set from England and Wales and is based on a 3-step analysis incl. (1) classification of communities, (2) transition model to describe annual changes in rivers communities and (3) water-quality-credit test to estimate environmental lags. The main manuscript is accompanied by extensive supplementary material which was only partly reviewed. I suggest separating the supplementary material in a part that is essential to see the methods of the main manuscript and another part that shows the details or tests the various methods, e.g. the different clustering methods.

My main general comments would be

- to critical question the "credit-debt-approach" and the idea that climate change (or any other anthropogenic stress) can (or should) be offset.
- Discuss the potential interaction of environmental variables.
- Warming in rivers may have different sources and one – cooling from power generation – may be significantly reduced in the future. What is actually the proportion of climate change warming and other warming and wouldn't be a much larger offset-effect visible from expected "re-cooling"?
- Observed / constructed conditions: I am actually not fully convinced that the approach works for the temperature data set – this seems not a significant trend in the observed values (Figure 4) and interestingly, the reconstructed trend has a different orientation, i.e. decreasing. What are potential explanations for this?

Furthermore the temperature / chemistry data set seems really disappointing (monthly values at sampling time) – this is also very different than for the hydrological data set. Last, I would argue that the mean distance between biological sites and chemistry/hydrology is not really short (mean 3.8 km) and even if on the main river course substantial changes may occur in case of tributaries which may render the interpolation not very correct.

Some detailed comments, per line

L39: before the "extinction" there should be a "population decline", or maybe "eventually become extinct".

L67/68: Somehow I am missing the link between these two lines (temperature and pollution)

L71: what is the difference between the "nearly 20000 samples" mentioned in the abstract and the " ~26000 samples" here?

L76: RE the first step, i.e. the classification: why did you create your own system and not results from an established assessment?

L100: did the same sites develop continuously? Somehow I lack the information which sites did transition and whether this was associated to some characteristics of space or changing environmental variables

L109: this paragraph is very different than the remain text and I am wondering why it was put here; also the comparison to the "tidal communities" and the mentioning of the resilient communities comes by surprise, as is the reference to the clustering method.

L136: this minimal environmental lag means there was no debt? I am a bit confused, because in L144 it says there may be debts and credits that may have cancelled each other out – which I find quite plausible.

Figures

Figure 1: Class 1 and 2 seem very similar in their overall prevalence development. Explained?

Figure 2: Describe what the colored arrows on top of each panel describe. I don't understand which data are shown – this seems to be a subset of the 3067 sites?

Figure 3: I would better frame this figure, including a map of Great Britain, a north arrow and some sort of scale. I am not entirely sure what is shown – these are the original class scores for the sites displayed? Why are there less sites from year to year?

Figure 4: Which points are shown in part d, e, f? Seems quite a small subset for such an ambitious analysis (but I may not grasp the full extent here).

Reviewer #2:

Remarks to the Author:

General comments:

This study by Vaughan & Gotelli addresses an important threat to freshwater biodiversity. Their study shows convincingly that even a very modest warming of 0.5 C will have important repercussions for macro-invertebrate communities.

Given that even diurnal fluctuations in water temperature will surpass 0.5 C, the insidious effects of chronic warming are underestimated in freshwater ecology, and effects of episodic heat waves or peak discharges have been emphasized in connection to climate change.

The potentially dramatic effects of a 0.5 C increase do not bode well for the 2C warming limit that we are currently headed for (and will likely surpass). The fact that dramatic changes have not been observed is because of parallel improvements in water. The authors arrive at this conclusion using a unique dataset, that has been previously utilized to answer questions of long-term change. The notion that improvement in local conditions may mitigate large scale climatic warming has been suggested made. However, what makes this study novel and unique is that it is the first in *quantifying* the expected and observed changes *separately* for temperature and water quality and for the *whole community*, demonstrating the validity for this concept.

They also juxtapose climatic 'debt' and environmental 'credit', which I have not seen before and which I think is a really intuitive way to communicate the problem of warming and the (partial) solution of water quality improvements to the general public.

Although I am not an expert on all the statistical analyses employed, I deem them appropriate and found the manuscript to be very accessibly written, the many different analyses and statistical details notwithstanding.

Detailed comments.

I have annotated the word file with (minor) suggestions which may help the authors in further improving their paper.

The strong correlation between reconstructed BOD and temperature was initially puzzling given that one would not expect these to be correlated: BOD is a proxy for the rate of oxygen consumption measured at a standard temperature. The result of BOD (i.e. water deoxygenation) would be expected to correlate with temperature, which it does (see the supplementary information in ref 24 where thermal effects are much stronger for levels of dissolved oxygen than they are for BOD). I then read in the supplementary material that the correlation between reconstructed BOD and reconstructed temperature actually reflects a similar biological response to either increasing temperature or to increasing BOD. This makes a lot of sense and actually strengthens the message of this study that 1) effects of warming can be mitigated by water quality improvements and 2) supports the assertion that using samples from the period 1991-1992 (when there was no correlation evident) for the transfer function is unlikely to introduce a bias.

Perhaps the link between BOD (a rate) and temperature (controlling this rate) could be better stressed. This also means that detrimental effects of warming are exacerbated under conditions of high BOD, where warming not only increases the oxygen demand of the organism, but also reduces the oxygen available to them (see ref 24).

Signed, Wilco Verberk

**Response to reviewer comments for manuscript NCOMMS-18-30291-T:
'Environmental credit paid the 20-year climatic debt of riverine invertebrate communities' (I.P. Vaughan & N.J. Gotelli)**

Notes:

- All references to pages and line numbers in this document refer to the original submitted version, in the pdf format sent for review.
- All changes described here are highlighted in red font in the main manuscript and supplementary material.

Reviewer #1 comments:

1. "I suggest separating the supplementary material in a part that is essential to see the methods of the main manuscript and another part that shows the details or tests the various methods, e.g. the different clustering methods."

We have re-formatted the manuscript to meet the *Nature Communications* requirements, with the Methods immediately after the Discussion. Following the journal's guidelines, we have kept the majority of the methods in the main paper. However, the detailed methods for the loglinear analysis, which are not needed to understand the main paper, have been moved to Supplementary Material, with only a brief description in the Methods.

2. "Critical[ly] question the "credit-debt-approach" and the idea that climate change (or any other anthropogenic stress) can (or should) be offset."

We have added two paragraphs to the start of the Discussion critically discussing offsetting, the credit-debt framework and interaction effects.

3. "Discuss the potential interaction of environmental variables."

Added to the start of the Discussion (see point 2 above).

4. "Warming in rivers may have different sources and one – cooling from power generation – may be significantly reduced in the future. What is actually the proportion of climate change warming and other warming and wouldn't be a much larger offset-effect visible from expected "re-cooling"?"

This is an interesting idea, and we are not aware of any study mapping the extent of thermal discharges and the downstream effects on river water temperatures at a national scale, in England and Wales, or any other country i.e. the contribution of power stations to overall water temperatures. However, the effects on our study system (either during the study period, or looking into the future) are likely to be small for two main reasons: i) there are few large

inland power stations in England and Wales (many of them are coastal; United Kingdom Statistics Authority (2018)) and ii) the inland ones tend to be restricted to larger rivers where the discharge is sufficient to meet their demands, generally excluding the wadeable streams in our data set (e.g. on the River Thames; Wright *et al.* 2000). As a consequence, the changing energy generation mix in future is unlikely to have a major effect on this system.

References:

United Kingdom Statistics Authority (2018) *Digest of United Kingdom Energy Statistics 2018*. Department for Business, Energy and Industrial Strategy.

Wright, J.F. *et al.* (2000) Minor local effects of a River Thames power station on the macroinvertebrate fauna. *Regulated Rivers: Research and Management*, **16**, 159-174.

5. “Observed / constructed conditions: I am actually not fully convinced that the approach works for the temperature data set – this seems not a significant trend in the observed values (Figure 4) and interestingly, the reconstructed trend has a different orientation, i.e. decreasing. What are potential explanations for this?”

The trend is significant, as shown in Supplementary Table 5, but this was not referred to in the figure legend. In response to this, we have: i) added reference to Supplementary Table 5 in Fig. 4’s legend, and ii) added two additional references to this table in the relevant part of the Results section.

The decrease in reconstructed temperature is simply a consequence of the large improvement in water quality: taxa sensitive to poor water quality and/or higher temperatures increased in prevalence, such that when the invertebrate community was used to reconstruct temperature, it suggested a slight cooling. We have added a point to clarify this in the Results:

‘Over the monitoring period, observed mean water temperature increased, whereas reconstructed temperature decreased – consistent with the observed increase in the prevalence of taxa preferring cooler conditions¹⁹ (i.e. Class 1)...’

6. “Furthermore the temperature / chemistry data set seems really disappointing (monthly values at sampling time) – this is also very different than for the hydrological data set.”

We agree that it would be nice to have temperature and water quality data at a higher temporal resolution (e.g. daily, like the hydrology data). However, we don’t think this is a serious problem given that we used annual median values to relate these variables to the annual macroinvertebrate data. Moreover, more frequent sampling across this extensive geographical network (>10,000 locations used for water chemistry and temperature; see Methods for details) would undoubtedly have resulted in fewer locations being sampled, increasing the average distance between temperature/chemistry and macroinvertebrate sampling locations.

7. “Last, I would argue that the mean distance between biological sites and chemistry/hydrology is not really short (mean 3.8 km) and even if on the main river course substantial changes may occur in case of tributaries which may render the interpolation not very correct.”

We agree that having a fully matched data set, with chemistry, hydrology and invertebrates recorded at the same locations is desirable, but unfortunately these are recorded in separate monitoring schemes. Similarly, tributaries could mean that chemistry changes substantially over short distances and amongst the >10,000 water chemistry locations used for the interpolation, this is likely to have occurred in some instances. However, this would merely represent ‘noise’ in our data, rather than a systematic bias. At most it will weaken the fit of our models used to reconstruct climate and water quality, but will not affect the overall results or the conclusions that we can draw from them.

Some detailed comments, per line

“L39: before the “extinction” there should be a “population decline”, or maybe “eventually become extinct”.”

We have expanded this to: ‘...or decline in abundance ultimately to extinction’

“L67/68: Somehow I am missing the link between these two lines (temperature and pollution)”

We have edited the text in this paragraph to tie these two concepts together more clearly. The relevant part of the paragraph now says:

“Riverine macroinvertebrates are sensitive to climate¹³, and water temperature increases equivalent to those observed in the current study often account for large changes in community structure (e.g. 14, 15). Concomitantly, as in much of Western Europe, water quality in British rivers has improved greatly in recent decades due to better treatment and reduced discharge of industrial and sewage effluent¹⁶. This has prompted large-scale biological recovery that may have offset concurrent water temperature increases¹⁷⁻¹⁹ (Supplementary Fig. 1).”

“L71: what is the difference between the “nearly 20000 samples” mentioned in the abstract and the “~26000 samples” here?”

This reflects differences in the final sample size used for different analyses, of which full details are given in the Methods. To avoid confusion, we have rephrased this sentence to say:

“Across a series of analyses using ~20–26,000 high-quality, standardized benthic macroinvertebrate samples...”

“L76: RE the first step, i.e. the classification: why did you create your own system and not results from an established assessment?”

Our aims here were twofold: i) to classify the data in multiple ways, varying the number of classes and clustering methodology, to ensure that our Markov chain analysis was robust to the choice of classification system, and ii) to show how a simple, generic classification approach can be combined with Markov chain analysis to analyse community dynamics. Established assessments would not provide that flexibility, nor make it so clear that only a simple classification is needed. In addition, water quality assessments usually entail taxon weightings based on their pollution sensitivity or similar, which would add an additional layer of complexity.

“L100: did the same sites develop continuously? Somehow I lack the information which sites did transition and whether this was associated to some characteristics of space or changing environmental variables”

Our aim in the paper was to look at system-wide properties, rather than the more detailed breakdown the reviewer suggests – although both are interesting questions. This is partially covered by the loglinear analysis of the transition probabilities, where we demonstrated that urban and rural rivers, and upland and lowland rivers, were showing similar patterns of transition probabilities through time. In the sentence following the one that the reviewer highlights, we describe the trends in transition probabilities through time among the biological classes. It would be possible to go into greater detail, but this would require a series of extra analyses (e.g. modelling transition probabilities among biological classes as a function of environmental variables) that we think would distract from the main focus of the paper.

We have edited the wording here to emphasise that our interest was the overall dynamics, rather than a detailed analysis of the factors affecting transition probabilities:

“Macroinvertebrate communities were dynamic, with approximately 20% of communities switching classes annually (Supplementary Fig. 5a; Supplementary Table 3) and long-term trends in the transition probabilities were consistent with national-scale improvements in water quality¹⁸...”

“L109: this paragraph is very different than the remain text and I am wondering why it was put here; also the comparison to the “tidal communities” and the mentioning of the resilient communities comes by surprise, as is the reference to the clustering method.”

We agree that there was little by way of introduction to this paragraph, so have added more explanation at the start of the previous paragraph where we introduce the Markov chain model. This now says:

“The classification was used to construct a time-varying Markov transition model that described annual changes in river community types through time (Fig. 2) and then allowed us to assess whether the dynamical stability of macroinvertebrate communities had changed during the study period.”

We have also added more details to the paragraph the reviewer highlights, clarifying why the comparison was made to the sub-tidal community, explaining resilience and enlarging on relevance of the clustering method here. In particular, we now say:

“There are few published examples of similar analyses, but these recovery rates are comparable to those estimated for sub-tidal benthic communities²⁰ and indicate a resilient system that can recover quickly from perturbation or respond rapidly to environmental change. These stability results were not sensitive to how the invertebrate communities were clustered prior to fitting the Markov chain model (Supplementary Fig. 5b).”

“L136: this minimal environmental lag means there was no debt? I am a bit confused, because in L144 it says there may be debts and credits that may have cancelled each other out – which I find quite plausible.”

Obviously the clarity of the wording could be improved here, to make it clear that adding together a similar sized debt and credit would lead to minimal net overall lag. We have clarified this in the second paragraph (formerly L144), so that the sentence now says:

“Although the Markov model revealed minimal net environmental lag in English and Welsh rivers, this does not preclude the formation of debts and credits that may have cancelled each other out, leading to a negligible net lag”

“Figure 1: Class 1 and 2 seem very similar in their overall prevalence development. Explained?”

We have discussed the increases in Classes 1 and 2 in more detail in the results, both in terms of the increase in overall prevalence (Fig. 1) and what this represents geographically (Fig. 3 and Suppl. Fig 2a). This is in the first paragraph of the ‘Markov chain modelling’ section of the Results. The second half of this paragraph now says:

“The prevalence of Classes 1 and 2 increased at similar rates (Fig. 1), but slightly faster for Class 2, driven by increasing probabilities through time of switching from Class to 3 to Class 2, decreasing probabilities of switching from Class 2 to 1, and increased persistence of communities within Class 2 (Fig. 2). Classes 1 and 2 expanded their geographic distributions, whilst Class 3 retreated mainly to urban areas and slow-flowing rivers in the

intensively farmed landscapes of eastern England (Fig 3; Supplementary Fig. 2a). Class 1 communities were concentrated in northern and western areas, where fast-flowing, well-oxygenated rivers are most abundant¹⁸, whereas Class 2 expanded across the lowlands of England (Fig. 3; Supplementary Fig. 2a)."

"Figure 2: Describe what the colored arrows on top of each panel describe. I don't understand which data are shown – this seems to be a subset of the 3067 sites?"

We have extended the first sentence of the figure caption to clarify this:

"Annual transition probabilities between the three macroinvertebrate community classes (1991–2011) across England and Wales, based on the Markov chain"

and added a further explanation:

"The fill colours of the arrows indicate the transitions between the different states"

"Figure 3: I would better frame this figure, including a map of Great Britain, a north arrow and some sort of scale. I am not entirely sure what is shown – these are the original class scores for the sites displayed? Why are there less sites from year to year?"

We have revised the figure in this way, adding a map of Great Britain with north arrow and scale bar, and with England and Wales shaded to clarify the study area. We have also reworded the legend for great clarity: the first part now says:

"Figure 3. The study area and locations of macroinvertebrate samples collected for calculating transition probabilities among years. (a) Outline of Great Britain, indicating England and Wales (shaded): the scale bar is 200km. (b) The samples collected in each year (sample sizes are given in Supplementary Table 1), shaded according the biological class to which they belonged."

"Figure 4: Which points are shown in part d, e, f? Seems quite a small subset for such an ambitious analysis (but I may not grasp the full extent here)."

Our focus here was on the credit and debt at the whole England and Wales scale: hence the relatively small number of points. Although mentioned in the opening line of the figure caption, we have reiterated this further down too, after describing panels d and e, by specifying that these are estimates from *"across England and Wales"*.

Reviewer #2 comments:

1. “I have annotated the word file with (minor) suggestions which may help the authors in further improving their paper.”

We have worked through this list and implemented all of the changes, except:

- Title of the paper. The reviewer makes a good point about not specifying England and Wales to reflect the wider relevance of the study. We have followed their advice, but re-arranged their wording slightly.
- L42 (first paragraph of the Introduction), adding “...or extinction debt”. We did not add this as it could be argued that an environmental lag could be too small to cause extinction, so extinction debt may not always be an appropriate alternative. In addition, the reference we cite here only talks about environmental lags.
- L55-8 (paragraphs 2-3 of the Introduction). The reviewer suggests a re-wording beginning with: ‘When adopting a broader perspective beyond temperature...’. We think this makes the explanation less clear, so we have not made this change, but instead have edited our own wording a bit to enhance overall clarity.
- Page 9: Added two of the four possible example papers suggested.

2. “The strong correlation between reconstructed BOD and temperature was initially puzzling given that one would not expect these to be correlated: BOD is a proxy for the rate of oxygen consumption measured at a standard temperature. The result of BOD (i.e. water deoxygenation) would be expected to correlate with temperature, which it does... I then read in the supplementary material that the correlation between reconstructed BOD and reconstructed temperature actually reflects a similar biological response to either increasing temperature or to increasing BOD. This makes a lot of sense and actually strengthens the message of this study that 1) effects of warming can be mitigated by water quality improvements and 2) supports the assertion that using samples from the period 1991-1992 (when there was no correlation evident) for the transfer function is unlikely to introduce a bias. Perhaps the link between BOD (a rate) and temperature (controlling this rate) could be better stressed. This also means that detrimental effects of warming are exacerbated under conditions of high BOD, where warming not only increases the oxygen demand of the organism, but also reduces the oxygen available to them (see ref 24).”

On the basis of the reviewer’s comment, we have moved the point about the observed temperature and BOD being weakly correlated, but their reconstructed equivalents being highly correlated, from the Methods into the final paragraph of the Results to strengthen the overall message.

Reviewers' Comments:

Reviewer #1:

Remarks to the Author:

Thank you for considering the remarks and comments!

Reviewer #2:

Remarks to the Author:

I believe that the points raised in the previous round of review have been satisfactorily addressed.

Wilco Verberk